# Real-time optical and electronic sensing with a β-amino enone linked, triazine-containing 2D covalent organic framework

Ranjit Kulkarni [1,2], Yu Noda [1,2], Deepak Kumar Barange [2], Yaroslav S. Kochergin[1,2,3], Pengbo Lyu [4], Barbora Balcarova[2], Petr Nachtigall[4] & Michael J. Bojdys [1,2]

Fully-aromatic, two-dimensional covalent organic frameworks (2D COFs) are hailed as candidates for electronic and optical devices, yet to-date few applications emerged that make genuine use of their rational, predictive design principles and permanent pore structure. Here, we present a 2D COF made up of chemoresistant β-amino enone bridges and Lewis-basic triazine moieties that exhibits a dramatic real-time response in the visible spectrum and an increase in bulk conductivity by two orders of magnitude to a chemical trigger - corrosive HCl vapours. The optical and electronic response is fully reversible using a chemical switch ($NH_3$ vapours) or physical triggers (temperature or vacuum). These findings demonstrate a useful application of fully-aromatic 2D COFs as real-time responsive chemosensors and switches.

[1] Department of Chemistry, Humboldt-Universität zu Berlin, Brook-Taylor-Str. 2, 12489 Berlin, Germany. [2] Institute of Organic Chemistry and Biochemistry of the CAS, Flemingovo nám. 2, Prague 166 10, Czech Republic. [3] Department of Organic Chemistry, Charles University in Prague, Hlavova 8, 128 40 Prague, Czech Republic. [4] Department of Physical and Macromolecular Chemistry, Charles University in Prague, Hlavova 8, Prague 128 40, Czech Republic. Correspondence and requests for materials should be addressed to M.J.B. (email: m.j.bojdys.02@cantab.net)

Organic semiconductor materials find use in sensing applications[1], opto-electronic devices[2,3], field-effect transistors[4], light-emitting diodes (LEDs)[5], photovoltaic and solar cells[6]. The efficiency of these devices largely depends on the band gap position and energy and charge-carrier mobilities. Band gap energy and charge transport depend not only on the degree of extended π-conjugation and planarity of the conjugated sub-units, but also on their chemical structure[7,8]. In contrast to many post-synthetic strategies for enhancing charge-carrier mobilities and band gap tuning, organic framework materials, such as covalent organic frameworks (COFs), offer predictive synthetic tools to design structure and properties from molecular building block to the extended macroscopic material. Thus, rational incorporation of electron-donating and with-drawing moieties into covalent organic frameworks can poten-tially improve their performance in electronic applications[9].

Covalent organic frameworks (COFs) are a well-established class of porous, crystalline networks made up from light elements (H, B, C, N and O)[10,11]. They are comprised of 2D covalently linked sheets that stack via π–π interactions. Early COFs are linked by boronate esters and boroxins that are easily hydrolysed and oxidised[12–15], and by more stable imines[16]. Their use in sensing applications and electrochemical devices has been ham-pered by the poor stability of these early COFs. We recently reported triazine ($C_3N_3$)-based graphdiyne frameworks that are not only crystalline but also stable with respect to a wide range of chemicals and temperatures and that show interesting photo-catalytic and electronic effects as well as structural polymorphism[17,18].

In this study, we condense a $C_2$ symmetric keto-enol, (1,4-phenylene)bis(3-hydroxyprop-2-en-1-one) (PBHP), and a $C_3$ symmetric tri-amine, 1,3,5-tris-(4-aminophenyl) triazine (TAPT), into a 2D-layered COF linked by β-amino enones (Fig. 1), in analogy to hydrolytically stable β-keto-enamine linkers obtained from $C_3$ symmetric keto-enols and $C_2$ symmetric amines[19]. This material is the first COF to show a rapid sensing response in the visible spectrum for volatile acid vapours, which we attribute to protonation of its Lewis-basic triazine-moieties accompanying with enhanced conductivity by ~170-fold in the protonated state. These effects take advantage of the open pore structure of the COF as well as its modular make-up that allows incorporation of Lewis-acidic and basic sites and, hence, enables intrinsic donor-acceptor behaviour that enhances band gap modularity and charge-carrier mobility. Intriguingly, in addition to the poly-crystalline powder morphology observed for most COFs, PBHP-TAPT COF grows in macroscopic, crystalline films at all flat interfaces of the reactor setup analogous to the surface-templated syntheses we used successfully for other layered 2D materials[20,21]. This unlocks applications in large-scale sensors and electronic devices, which we explore herein.

## Results

### β-amino enone linked, triazine-containing 2D covalent organic framework.

The triazine-containing PBHP-TAPT COF is an extension of the family of 2D COFs obtained via Michael addition-elimination[22]. The triazine-moiety in particular is an intriguing building block for frameworks with electronic and sensing end-applications, since its electron-accepting capacity is enhanced in the excited, protonated state[23]. Formation of the β-amino enone bridge is accompanied by irreversible tautomer-ization[24], which is likely to produce defects and, hence, a dis-ordered framework under standard conditions. Hence, we condensed PBHP and TAPT in a 3:2 molar ratio under sol-vothermal conditions in a sealed glass ampule to allow for dynamic bond formation close to equilibrium (Fig. 1). PBHP-

TAPT COF precipitates as an orange solid and grows on all reactor interfaces as an orange film with an overall yield of 79% (Supplementary Fig. 3, for detailed procedures). We optimized the synthetic protocol by monitoring the intensity of crystalline peaks in powder X-ray diffraction (PXRD) measurements with varying ratios of the solvent mixture, mesitylene and dioxane[25,26]. The highest intensity of characteristic peaks was obtained for a 9:1 volume ratio of mesitylene to dioxane (Supplementary Fig. 4).

Combustion elemental analysis of PBHP-TAPT COF shows a composition of C, H and N at 71.3%, 4.4% and 12.5%, respectively, which is reasonably close to the theoretical values of 74.6%, 4.3% and 13.3% (Supplementary Table 1). Thermo-gravimetric analysis (TGA) under air and nitrogen shows that PBHP-TAPT COF is stable up to ~420 °C. Since no appreciable weight loss is observed before the onset of decomposition, we conclude that the framework is virtually free of trapped guest molecules (Supplementary Fig. 5). The FT-IR spectra of PBHP-TAPT COF powder show the disappearance of N–H stretches and the emergence C–N and C=N vibrations at 1223 and 1628 cm$^{-1}$ (Fig. 2a, Supplementary Fig. 22)[22,27]. Likewise, O–H vibrations seen for the keto-enol PBHP between 2800 and 2900 cm$^{-1}$ disappear completely after condensation. Importantly, characteristic signals of the TAPT building blocks stemming from the triazine stretching mode (1501 cm$^{-1}$), breathing mode (1366 cm$^{-1}$) and out-of-plane ring bending (809 cm$^{-1}$) are retained[18]. $^{13}$C cross-polarization magic angle spinning (CP-MAS) solid-state NMR spectroscopy confirms the formation of β-amino enone bridges showing characteristic signals for the carbonyl carbons at 189 ppm, the β-carbon at 142 ppm, and the α-carbon at 95 ppm (Fig. 2b)[22]. Triazine ring carbon environ-ments are assigned to the peak at 169 ppm[21].

The PXRD pattern of PBHP-TAPT COF confirms the formation of a 2D-layered hexagonal network. Indexing and Le Bail refinement readily matches the observed profile using a hexagonal unit cell P-6 (no. 174) ($R_{wp}$ = 1.10%). To account for the stacking disorder we modelled conceivable stacking modes using density functional theory (DFT). Three arrangements typical for 2D COFs were considered: AA-eclipsed, AA'-serrated and AB-staggered. Calculations show further that the AA and AA' packing modes are significantly more stable than the AB-staggered structure (Supplementary Table 2). Indeed, the AA packing mode gave the best match with the observed diffraction profile (Fig. 2c, d, and Supplementary Figs. 6 and 7), and it was subsequently used for structure refinement using Pawley methods with a low residual ($R_{wp}$ = 3.91%). PBHP-TAPT COF has a dominant diffraction peak at 2.13° as well as smaller peaks at 3.8°, 4.5° and 6.1° and a broad peak at 25.5° 2θ (Cu Kα λ = 0.15406 nm). These peaks correspond to the (100), (110), (200), (210) in-plane reflections as well as the (001) stacking reflection. In-plane distances between neighbouring pore chan-nels are 4.1 nm, hence, the expected pore diameter is ~2.0 nm, and the average inter-layer distance is 0.34 nm, which is typical for van der Waals stacked 2D materials.

Scanning electron microscopy (SEM) shows that the bulk of PBHP-TAPT COF powder consists of spherical particles with diameters ranging between 5 and 20 μm, which is indicative of emulsion polymerisations (Supplementary Fig. 8). However, the powder can be mechanically broken up using ultrasonication, and reveal flat terraces in atomic force microscopy (AFM) measure-ments with average flake thicknesses of 9.9 nm (Supplementary Fig. 9). High-resolution transmission electron microscopy (HR-TEM) confirms the inter-layer spacing as 3.6 Å (Supplementary Fig. 10).

Nitrogen sorption isotherms of PBHP-TAPT COF show Type I behaviour, indicative of pores, with a guest-accessible Brunauer–Emmett–Teller surface area ($S_{BET}$) of 176 m$^2$ g$^{-1}$

**Fig. 1** Reaction scheme for PBHP-TAPT COF and its constituent building blocks

(Supplementary Fig. 11d). Pore size distribution was calculated using quenched solid-state functional theory (QSDFT) (Supplementary Fig. 11e), and the largest contribution to pore volume stems from pores of diameter 3.9 nm, which is close to the pore channel diameter of the refined structure of 3.6 nm (Supplementary Fig. 11a). Note, that the observed surface area is far from the expected Connolly surface area of 2399 $m^2\,g^{-1}$. Since no evidence of stoichiometric incorporation of guest molecules was found by elemental analysis, [13]C CP-MAS solid-state NMR, and TGA, we conclude that stacking defects or interdigitation of neighbouring layers must account for the apparent loss in accessible surface area. Indeed, the calculated energy differences between stacking

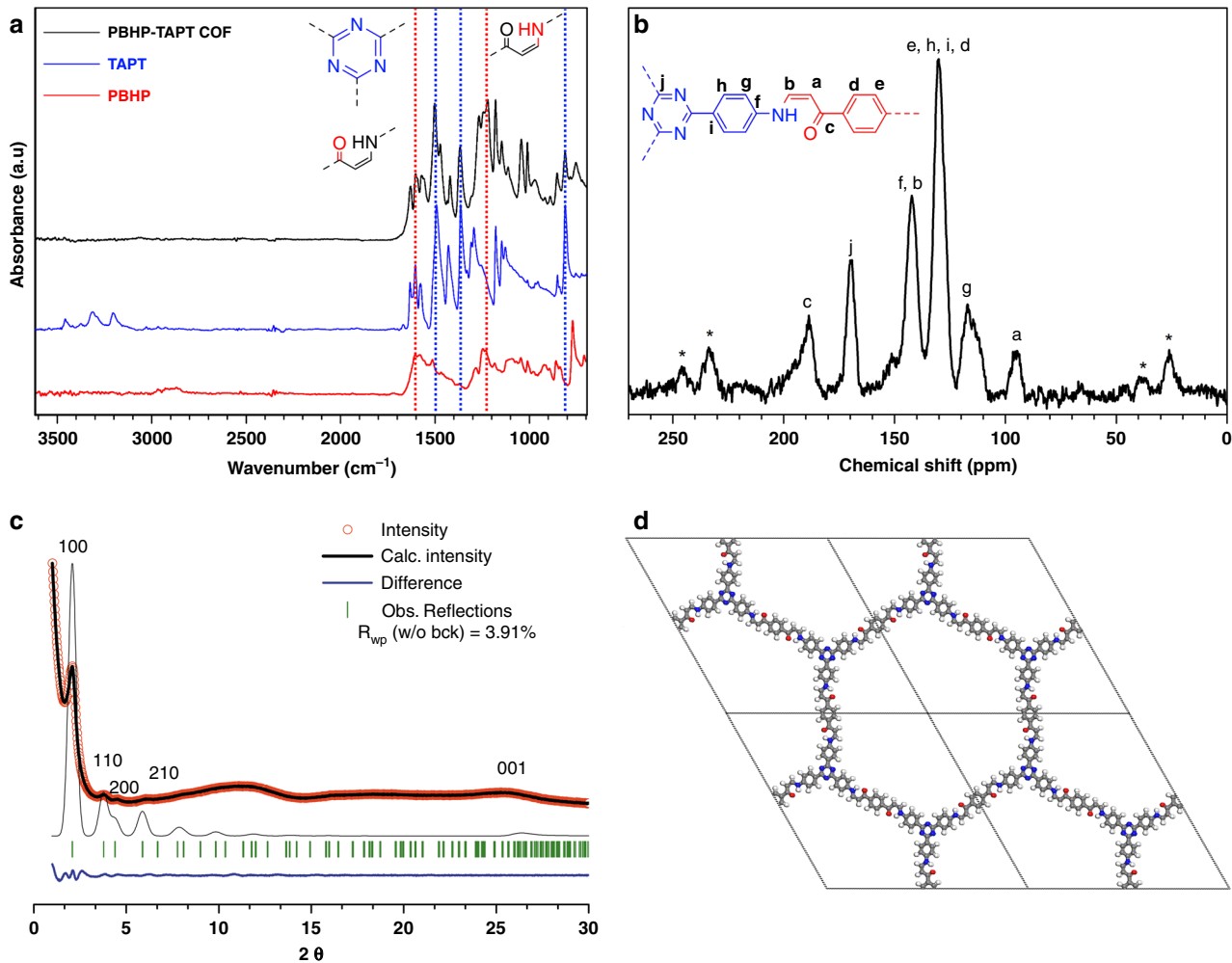

**Fig. 2** Physical characterization of PBHP-TAPT COF. **a** Comparison of the Fourier-transform infrared (FT-IR) spectra of PBHP-TAPT COF (in black) with its building blocks TAPT (in blue) and the PBHP (in red). **b** $^{13}$C cross-polarisation (CP) magic angle spinning (MAS) NMR (MAS rate of 13 kHz) of PBHP-TAPT COF. Spinning sidebands are marked with asterisks (*). **c** Experimental (red circles), Pawley refined (thick black line), and predicted (thin black) PXRD patterns of PBHP-TAPT COF. Bragg peak positions (green), and the difference plot (experimental minus refined; blue). **d** 2 × 2 unit cells (a = 44.2785 Å, c = 3.3767 Å, space group 174) of PBHP-TAPT COF with the principal unit cell (black dashes). Carbon (grey), hydrogen (white), nitrogen (blue), oxygen (red) atoms are represented as spheres

modes vary between 0.1 and 4.3 eV (Supplementary Table 2), hence, there is a high likelihood that even a small number of stacking defects occlude most of the theoretically accessible surface area.

The modest surface area of PBHP-TAPT COF was, however, achieved without the need for elaborate post-synthetic activation techniques such as supercritical $CO_2$ or solvent exchange which find their way into COF literature (Supplementary Fig. 11d)[22,28].

The orange colour of PBHP-TAPT COF stems from an absorption edge at 525 nm according to solid-state UV/Vis diffuse reflectance spectroscopy (Supplementary Fig. 12), and according to the Kubelka-Munk function it corresponds to a direct band gap of 2.32 eV and an indirect band gap of 2.06 eV. Solid-state photoluminescence (PL) spectrum shows an emission maximum at 590 nm (2.10 eV), which is closer to the calculated indirect band gap value (Supplementary Fig. 13), suggesting that PBHP-TAPT COF is an indirect band gap semiconductor. This combination of an optical band gap in the visible spectrum, guest-accessible pore volume, a strong covalent backbone and the presence of conjugated Lewis-basic moieties in the form of

triazine cores make PBHP-TAPT COF an excellent candidate material for sensing of volatile, corrosive gases.

**Real-time opto-electronic response.** In a first set of experiments, we tested the response of PBHP-TAPT COF to HCl vapours. The as-received powder showed a rapid colour change from orange to red (shift of the UV/Vis absorption edge to 630 nm) within seconds of exposure to HCl vapours at 1 atm and RT. This coloration is fully reversible when the HCl-treated sample is exposed to $NH_3$ vapours at the same conditions (shift of the UV/Vis absorption edge back to 525 nm) (Supplementary Fig. 14 and Supplementary Movie 1). The reversibility of the PBHP-TAPT COF was investigated by cycling the reaction with HCl and $NH_3$ probes. The cycling test showed the retention of sensing ability of PBHP-TAPT COF, in five consecutive HCl-$NH_3$ exposure cycles (Supplementary Fig. 15a). Structural integrity of PBHP-TAPT COF after cycles with HCl/$NH_3$ gas exposure was confirmed with Fourier-transform infrared (FT-IR) spectra measurements (Supplementary Fig. 15b). The cycled samples showed no marked changes in the skeleton of PBHP-TAPT COF upon exposure to HCl/$NH_3$ gas.

Furthermore, PBHP-TAPT COF, was highly sensitive to low concentration of HCl gas down to 20–50 ppm, detectable by UV-Vis. and the higher threshold was evaluated at around 3000 ppm (Supplementary Fig. 16). This rapid, colour change offers an instant advantage over other optical sensors for corrosive gases where good reversibility and real-time response are lacking[29,30].

In the protonated, excited state the direct and indirect band gaps of PBHP-TAPT COF decrease to 2.00 and 1.78 eV, respectively—a change by ~0.3 eV. PBHP-TAPT COF remains structurally stable throughout the exposure to HCl and $NH_3$ vapours as seen in PXRD profiles (Supplementary Fig. 17). Intriguingly, the protonated, excited framework can also be fully regenerated by thermal treatment (>120 °C for 60 min) or in vacuum. Structural integrity of PBHP-TAPT COF after physical and chemical regeneration was confirmed with Fourier-transform infrared (FT-IR) spectra and solid-state UV-Vis measurements (Supplementary Fig. 26).

The trend in colour change observed in UV/Vis is mirrored in PL spectroscopy. Not only does the emission maximum shift to 660 nm (1.84 eV) upon HCl exposure, but the overall fluorescence is also quenched. Calculations suggest that the triazine ring has a larger hyperpolarisability and more electron withdrawing character than the analogous carbon only benzene core[31]. We therefore conclude that the observed fluorescence quenching of protonated PBHP-TAPT COF is a consequence of electron donation into the protonated triazine ring, an effect that we have previously observed for strong donor-acceptor interactions in triazine-containing, conjugated microporous polymers[31] and for molecular compounds[32].

PBHP-TAPT COF has in principle three potential sites that can act as Lewis-bases (Supplementary Fig. 18), so to verify the site at which protonation occurs, we synthesised two sub-units of the COF backbone (Supplementary Fig. 18), keto-enamine (KE) and tri-phenyl triazine (TPT) and two model compounds 1,3,5-tris(4-aminophenyl)benzene (TAP) and 1,3,5-tris-(4-aminophenyl) triazine (TAPT) (Supplementary Fig. 19). These model compounds were exposed to HCl vapours at the same condition as the bulk powder and UV/Vis spectra were recorded. As expected, only the protonated, triazine-containing model compounds showed a red shift of the absorption edge, while the KE core did not (Supplementary Fig. 18b and 18c). The solid-state $^{13}C$ CP-MAS NMR spectrum of the protonated PBHP-TAPT COF has the best correlation with the triazine ring-nitrogen as the preferred protonation site (Supplementary Figs. 27 and 28). A computational investigation of the protonation of PBHP-TAPT COF shows that a protonated triazine linker is preferred by 70 kJ mol$^{-1}$ over a protonated keto-enamine bridge (Supplementary Figs. 20 and 21). The ring protonation of triazine core affecting the red shift have already been reported for some oligomeric/polymeric systems and our observation also complies with it[23,31,33,34]. Further evidence that the site of protonation is indeed the triazine core comes from FT-IR spectroscopy (Supplementary Fig. 22). None of the signals belonging to the β-amino enone bridges or the aromatic backbone of PBHP-TAPT COF see any appreciable changes upon exposure of the material to HCl vapours. However, the typical absorption peaks of the triazine units shift in the protonated network from 1501 to 1504 cm$^{-1}$ (stretching mode), from 1366 to 1362 cm$^{-1}$ (breathing mode) and from 809 to 811 cm$^{-1}$ (out-of-plane bending) These miniscule shifts have been reported previously for protonated triazines[32,35,36]. Peak positions are fully recovered after regeneration using $NH_3$ vapours.

Thus, we conclude that the basic triazine moieties are preferentially protonated when the framework is exposed to HCl vapours. This protonation gives the triazine moieties a localized positive charge, drawing π-electron density from its neighbouring functional groups[23]. The reversible optical and electronic response of the bulk powder is all-encompassing with no evidence of mixed-phase behaviour, which suggests that HCl (and later $NH_3$) vapours can access the entire latent pore volume. Spontaneous access to latent pore volume can be triggered by local structural changes and has been observed previously for 2D porous materials that were exposed to strongly interacting guest molecules such as $CO_2$ or iodine in contrast to the physisorption of weakly interacting guests such as $N_2$ and Ar[37,38].

We have seen so far that reversible activation of PBHP-TAPT COF by acid vapours not only decreases the optical band gap by 0.3 eV, but also attenuates the photoluminescence of the material. (Supplementary Fig. 13 and Supplementary Movie 2). Protonation of the triazine moieties is the most likely explanation for this behaviour, yet there is a composite effect that involves the entire aromatic backbone of PBHP-TAPT COF. Thus, we examine the effect of this reversible chemisorption on the charge-carrier mobility in bulk PBHP-TAPT COF (Supplementary Fig. 23). To date, the most common technique to tune the electrical conductivity of COFs is iodine or sulphur doping[39–41]. However, to the best of our knowledge there are no reports to reversibly tune charge-carrier mobility in COFs in real-time. Pristine PBHP-TAPT COF has a normalised conductivity of $1.32 \times 10^{-8}$ S m$^{-1}$. On protonation, the conductivity of activated PBHP-TAPT COF increases by 170-fold to $2.18 \times 10^{-6}$ S m$^{-1}$ (Fig. 3c). Furthermore, the conductivity drops close to the original value ($1.23 \times 10^{-8}$ S m$^{-1}$) when the sample is regenerated with $NH_3$ vapours. This trend in the overall electrical conductivity is reproduced up to five cycles (Supplementary Fig. 24).

In summary, we report a 2D COF made up of chemoresistant β-amino enone bridges (PBHP) and Lewis-basic triazine moieties (TAPT) capable of real-time, reversible optical and electronic sensing of volatile acids and bases. This study shows that optical and electronic activation of PBHP-TAPT COF is achieved by preferential protonation of the triazine nitrogens, which results in an optical response visible to the naked eye and an increase of bulk conductivity by two orders of magnitude. The single-site protonation triggers π-electron donation into the triazine ring and is accompanied by fluorescence quenching. Both the optical and electronic effects involve the entirety of the π-aromatic framework and are made possible because PBHP-TAPT COF is not only chemically stable to the corrosive trigger molecules, but it is also permanently porous to the chemisorbed guests. Activation of the framework is fully reversible by chemical triggers ($NH_3$), by heating or by vacuum. These findings demonstrate a powerful approach to design more practical sensors and switches, and take genuine advantage of the chemoresistant make-up, of the porous structure, and of the overall conjugation of fully-aromatic donor-acceptor PBHP-TAPT COF.

## Methods

**General**. Unless otherwise stated, all materials were commercially available and used without further purification. All solvents were of analytical grade and used without further purification. 4-Aminobenzonitrile, trifluoromethanesulfonic acid (TFMSA) were purchased from Acros Organics. 1,4-Diacetylbenzene, potassium tert-butoxide and ethyl formate were purchased from Sigma–Aldrich.Co. Ltd. (TCI).

**Characterization**. Powder X-ray (PXRD): measurements were performed with a Bruker D8 Advance diffractometer using Bruker AXS D8 Advanced SWAX diffractometer with Cu Kα (λ = 0.15406 nm) as a radiation source. Samples were measured from 1 to 80° 2θ with the step of 0.021° 2θ secondary graphite monochromator and LYNXEYE XE detector. Elemental analyses (EA) (C, H and N): were performed using a PE 2400 Series II CHN Analyzer. Fourier transformed infrared spectroscopy (FT-IR): spectra were recorded on an AVATAR 370 FT-IR

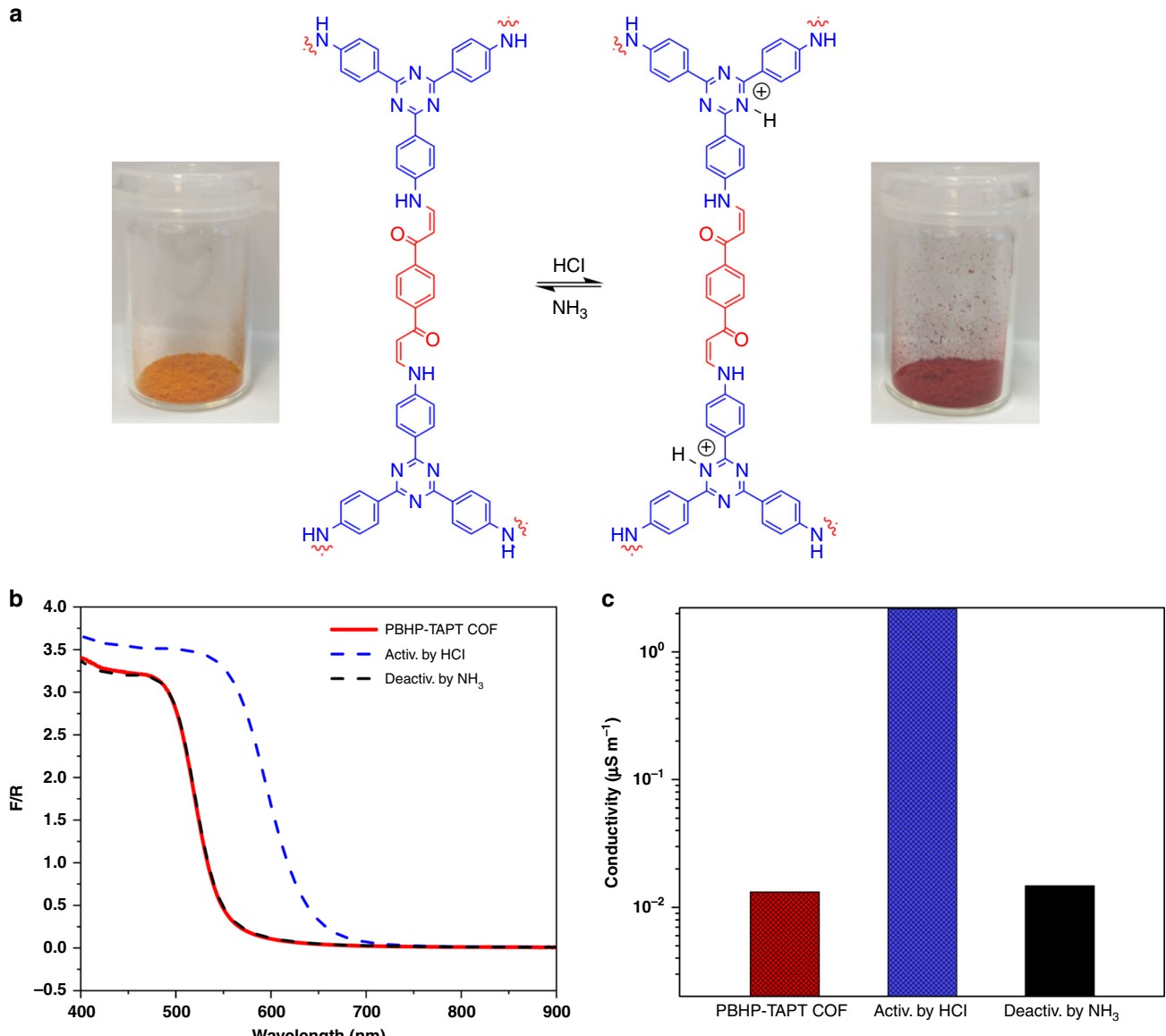

**Fig. 3** Opto-electronic response of PBHP-TAPT COF. **a** Protonation and deprotonation of the triazine moieties with chemical triggers (HCl and NH$_3$ vapours): Pristine, bulk PBHP-TAPT COF orange powder (left) and the activated, red powder, after 2–3 s of exposure to HCl (right). **b** Solid-state UV/Vis spectra of pristine PBHP-TAPT COF (red), after activation with HCl vapours (blue; dashed and dotted line), and regeneration with NH$_3$ vapours (black; dashed line). **c** Electrical conductivity of pristine PBHP-TAPT COF (red), after activation with HCl vapours (blue), and regeneration with NH$_3$ vapours (black)

spectrometer from Thermo Nicolet. Solid-state nuclear magnetic resonance (NMR) spectroscopy: $^{13}$C CP/MAS spectrum (NMR) of the TAPT-COFs was recorded on Agilent DD2 600 Solid NMR System with 3.2 mm zirconia rotors. The spinning rate is 13 kHz and the contact time was 10 ms. $^1$H and $^{13}$C NMR spectra of the monomers were recorded on a Bruker Advance 400 instrument. Chemical shifts (δ) were reported in ppm. Thermogravimetric analysis (TGA): Thermal gravimetric analysis was carried out on Setsys Evolution 18 thermal analyser from Setaram instrument by heating the samples from 50 to 800 °C under air or nitrogen atmosphere at a heating rate of 10 °C min$^{-1}$. Solid-state UV/Vis measurements (UV/Vis): were undertaken with a 6000i UV-Vis-NIR spectrometer from Agilent, and solid-state fluorescence measurements recorded on Fluorolog FL3-22 fluorometer (Horiba–Jobin Yvon). Gas sorption measurements and pore size distributions of COFs (BET): N$_2$ adsorption desorption isotherm was tested on Autosorb iQ instrument and measured using Helium mode. All Samples for nitrogen sorption at 77 K were activated by degassing in vacuum (2 × 10$^{-5}$ mbar) at 110 °C for 24 h. The surface area was calculated in the relative pressure (p/p0) range from 0.05 to 0.35. Pore size distributions were calculated for the adsorption as well as for the desorption branch using QS-DFT and NL-DFT models. Testing process was carried out in liquid nitrogen at 77 K and the specific surface area of the sample was calculated according to the BET (Brunauer–Emmett–Teller). Scanning electron microscope (SEM): images were

obtained with a Nova NanoSEM 450 from FEI. The dry samples were prepared on 15 mm aluminium stubs using an adhesive, high purity carbon tab. Imaging was conducted at a working distance of 5 mm and a working voltage of 3–10 kV using a mix of upper and lower secondary electron detectors. The field emission scanning electron microscope measurement scale bar was calibrated against certified standards. Transmission electron microscopy (TEM): was carried out using a The FEI Talos™ F200X is a 200 kV FEG operating at an accelerating voltage of 200 kV with a spherical aberration coefficient: value of 5.6 mm. Images were recorded on Ceta™ 16 M camera, which combined with an embedded Piezo-enhanced stage. Atomic force microscopy (AFM): measurements were performed with Bruker Multimode VIII equipped with E-Scanner. The PBHP-TAPT COF was suspended in THF and sonicated for 30 min in an ultrasonicator. Then the supernatant solution was drop casted onto a silicon substrate (with 300 nm-SiO$_2$ layer). Peak Force Tapping mode was employed with the cantilever SCANASYST-AIR.

**Conductivity measurements**. Electrical conductivity was determined by two-point I–V measurements. Powder samples were pelletized into a disk between two stainless rods supported by an insulating plastic insert with an inner diameter of 8 mm. I–V profiles were obtained and recorded at RT using a source meter, Keithley

2612 A, with the voltage ranging from −10 to 10 V in typical experiments. Average of minimum three separate measured current values was analysed to calculate the conductivity of each sample.

**Computational methods**. All the simulations are based on density functional theory and were performed using the projector augmented wave (PAW), formalism within the generalised gradient approximation (GGA) method with Perdew-Burke-Ernzerhof (PBE) exchange-correlation functional as implemented in Vienna Ab Initio Simulation Package (VASP). The cut-off energy of 800 eV for the plane-wave basis set has been consistently used in all calculations. The convergence criterion of 0.01 was used for the forces in geometry optimizations and $10^{-5}$ eV was used for the energy convergence. The lattice parameters and atomic positions were fully relaxed. The Brillouin zone for structure optimizations was sampled with the Monkhorst-Pack, special k-point mesh on $1 \times 1 \times 3$. D3 dispersion correction was adopted in all calculations. All geometry optimizations were performed with the unit cell consisting of two layers of COF, starting from the geometry described as AA-eclipsed, AA-serrated and AB. The total energies of AA-eclipsed, AA-serrated and AB are −2101.53, −2101.43 and −2097.23 eV, respectively. Both AA structures are more stable than AB.

## Data availability

All data generated or analysed during this study are provided as a Source Data file at https://doi.org/10.5281/zenodo.3249392 (URL: https://zenodo.org/record/3249392).

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

## Acknowledgements

This work was primarily funded by European Research Council (ERC) for funding under the Starting Grant Scheme (BEGMAT-678462). We thank Adam Malek for access to IR spectroscopy, and Dr. Martin Dracinsky for access to solid-state NMR facilities. Christoph Erdmann and Prof. Nicola Pinna for TEM measurements. Prof. Emil List Kratochvil for the access to their integrating sphere. Prof. Dr. Jürgen P. Rabe for access to AFM measurements and Prof. Christoph T. Koch and Johannes Müller for some additional SEM measurments.

## Author contributions

R.K. carried out the synthetic experiments, analysed the data and wrote the paper. Y.N. carried out the conductivity measurements and analysis. D.K.B. and R.K. synthesised the

sample compounds and building blocks. Y.S.K. helped with the solid-state UV/Vis and thermogravimetric analysis. B.B performed $N_2$ sorption measurements and analysis. P.L. and P.N. performed the DFT calculations. M.J.B. conceived the experiments and wrote the manuscript.

## Additional information

**Competing interests:** The authors declare no competing interests.

