## [Peer Review File · Nature Communications]

Reviewers' comments:

Reviewer #1 (Remarks to the Author):

In this paper, Bojdys and coworkers reported a triazine-based 2D covalent organic framework (PBHP-TAPT COF) synthesized through Michael Addition–Elimination. The authors claim that PBHP-TAPT COF can be utilized as a real-time responsive chemosensor to HCl vapor. Moreover, after the protonation, the conductivity of the COF increased by two orders of magnitude. This paper presents some interesting results, but the quality of the COF material is not good. Therefore, I do not see the current version presents results are novel enough and can be published in Nature Communications. Before resubmission, the author should consider the following points:

1. The crystallinity of this COF is not good. The author should optimize the reaction condition to improve the crystallinity of COF.
2. The BET surface area of this COF is 178 cm³/g, which is much lower to the theoretical value and should be the lowest values for a COF that I have heard. Usually, the low BET surface area is related to the poor crystallinity.
3. The PSD reveals several peaks for this COF, which is unreasonable. According to the crystal model, there should have only one peak.
4. For structural simulation, there is an obvious deference between the experimental pattern and simulated patterns for AA-Serrated. The author should do the simulation again.
5. The author used several techniques to verify the site for protonation. Did they try the solid-state NMR? There should some shift for the carbon of triazine ring during this protonation process.

Reviewer #2 (Remarks to the Author):

This manuscript reported the fabrication of a 2D COF from chemoresistant β -amino enone bridges and Lewis-basic triazine moieties. The obtained 2D COF material exhibits a rapid sensing response in the visible spectrum for volatile acid vapours. To date, a series of chemical reactions, synthetic methods and synthetic strategies have been developed, which greatly promoted the development of COFs. However, relative application of COFs materials in large-scale sensors is rare. This guarantee its publication in nature communications. Several issues should be addressed prior acceptance.

1. COFs materials usually have uniform pore size distribution. Why does this material have such a wide range of pore size distributions?
2. The author said "the framework can also be fully deactivated by a mild thermal treatment (> 120 °C for 60 min) or in vacuum". What is the meaning of deactivation? Whether the material still realize the reversible activation by acid vapours after such a treatment? What is its structural change after such a treatment?
3. From the manuscript, it can be seen that the pristine PBHP-TAPT COF has a normalised conductivity of $1.32 \times 10^{-8} \text{ S m}^{-1}$. On protonation, the conductivity of activated PBHP-TAPT COF increases to $2.18 \times 10^{-6} \text{ S m}^{-1}$. However, compared with the other materials the author listed, this conductivity is still low. What is the reason for the authors to choose 2Z,2'Z)-(1,1'-(1,4-phenylene)bis(3-hydroxyprop-2-en-1-one) instead of 1,4-diacetylbenzene as the starting material? If selected 1,4-diacetylbenzene, the formed COF material should have a fully π -conjugated framework, which in turn will have a better conductivity.

Reviewer #3 (Remarks to the Author):

It is an excellent manuscript on the topic of "A real-time optical and electronic chemical sensor based on a β -amino enone linked, triazine-containing 2D covalent organic framework ". I think the m/s has been written well and the logic behind the m/s is nice. Hence I recommend acceptance per minor revision.

1. Once I look into the structure I felt that there is a high chance of an intramolecular hydrogen

bonding [See *Angew. Chem. Int. Ed.*, 2013, 52, 13052–13056]. I was wondering if the authors saw some indication of such.

2. Please see the N-H next to b in the figure 2b. If we replace that N-H with N-D via a D exchange I guess the possibility of this hydrogen bonding will more clear.

3. Similarly, an interlayer hydrogen bonding could be another possibility as well. See *J. Am. Chem. Soc.*, 2018, 140, 10941

4. The free C=O functionality and the very existence of that indicate such possibilities.

Reviewer #1 (Remarks to the Author):

In this paper, Bojdys and coworkers reported a triazine-based 2D covalent organic framework (PBHP-TAPT COF) synthesized through Michael Addition–Elimination. The authors claim that PBHP-TAPT COF can be utilized as a real-time responsive chemosensor to HCl vapor. Moreover, after the protonation, the conductivity of the COF increased by two orders of magnitude. This paper presents some interesting results, but the quality of the COF material is not good. Therefore, I do not see the current version presents results are novel enough and can be published in Nature Communications. Before resubmission, the author should consider the following points:

1. The crystallinity of this COF is not good. The author should optimize the reaction condition to improve the crystallinity of COF.

We respectfully disagree with our colleague. Qualitative assessment of the crystallinity of COFs (“not good”) is somewhat ambiguous. Hence, we have chosen to compare the FWHM of the principle (100) peak in the PXRD profiles of our material with that of several examples from literature. The (100) peak of our reported PBHP-TAPT COF has a FWHM of 0.4°, while other COFs have similar FWHM of 0.4-0.7° (“Tuneable near white-emissive two-dimensional covalent organic frameworks” Nat. Commun. 2018, 9, 2335) and 0.39-0.6° (“Stable, crystalline, porous, covalent organic frameworks as a platform for chiral organocatalysts” Nature Chemistry, 2015, 7, 905–912). Evidently, our material has a degree of crystallinity comparable to other COF references in literature. In addition, I would like our colleague to note, that the currently reported synthetic protocol for PBHP-TAPT COF has been subject to extensive screening and optimisation (Supplementary Section 3 and 4, Supplementary Fig. 1).

2. The BET surface area of this COF is 178 cm³/g, which is much lower to the theoretical value and should be the lowest values for a COF that I have heard. Usually, the low BET surface area is related to the poor crystallinity.

Related reports of COFs obtained via Michael addition elimination methods all seem to show low accessible surface areas of roughly around 150-200 m²/g for example by Perepichka et al. (J. Am. Chem. Soc. 2017, 139, 2421-2427). In the context of the presented sensing application a higher N₂ BET surface area (post-activation) offers no tangible benefit.

Also note, that only small energy differences of 0.1 – 4.3 eV per unit/cell can lead to substantial changes of pore morphology (as we show in the newly prepared Supplementary Fig. 8) blocking up theoretically accessible pore space.

We have tried to address this point of the reviewer in the context of his question 3 below (and as follows on p.9):

“Note, that the observed surface area is far from the expected Connolly surface area of 2399 m² g⁻¹. Since no evidence of stoichiometric incorporation of guest molecules was found

by elemental analysis, ^{13}C CP-MAS solid-state NMR, and TGA, we conclude that stacking defects or interdigitation of neighbouring layers must account for the apparent loss in accessible surface area. Indeed, the calculated energy differences between stacking modes vary between 0.1 and 4.3 eV (Supplementary Table 2), hence, there is a high likelihood that even a small number of stacking defects occlude most of the theoretically accessible surface area.”

3. The PSD reveals several peaks for this COF, which is unreasonable. According to the crystal model, there should have only one peak.

According to the energetically close, possible packing modes of the COF structure (AA, AA' and AB) at least two different pore sizes are conceivable: pores with diameters of 3.6 nm (for AA, and AA') and 1.4 nm (for AB) (as we show in the newly prepared Supplementary Fig. 8). The newly added pore size analysis based on quenched solid-state functional theory (QSDFT) predicts that the largest contribution to pore volume stems from pores of diameter 3.9 nm which is close to the pore channel diameter of the refined structure (3.6 nm) (Supplementary Fig. 8a), with some contribution of pores with diameter of 2.1 nm; overall, a much better fit. While NLDFT is the preferred method for calculation of pore sizes in the materials community (and hence was included in the first place), we must remember that it is a method that works best for siliceous materials and zeolites: pore size analysis of carbons with polar surface functionalities and flexible pore structures remains difficult to predict with NLDFT. There is an excellent article that discusses the pros and cons of NLDFT and QSDFT elsewhere (J. Landers et al. / Colloids and Surfaces A: Physicochem. Eng. Aspects 437 (2013) 3–32). We chose to include results for PSD analysis from NLDFT and QSDFT methods in a re-worked Supplementary Fig. 8, especially, since QSDFT is becoming increasingly more common for COF materials and heterogeneous carbons. (Carbon . 2014, 72, 47-56) , (Colloids and Surfaces A: Physicochem. Eng. Aspects, 201, 437 3-32), (Chem. Mater. 2016, 28, 626–631), (Langmuir 2006, 22, 11171-11179), (CrystEngComm, 2016, 18, 4295–4302) and (Nat Commun 2018, 9,2600)

We have also amended the discussion of PSD analysis on p. 9 as follows:

“Nitrogen sorption isotherms of **PBHP-TAPT COF** show Type I behaviour, indicative of pores, with a guest-accessible Brunauer–Emmett–Teller surface area (S_{BET}) of $176 \text{ m}^2 \text{ g}^{-1}$ (Supplementary Fig. 8d). Pore size distribution was calculated using quenched solid-state functional theory (QSDFT) (Supplementary Fig. 8e), and the largest contribution to pore volume stems from pores of diameter 3.9 nm which is close to the pore channel diameter of the refined structure of 3.6 nm (Supplementary Fig. 8a).”

Figure Modified in the SI :

Supplementary Fig. 8: Calculated pore diameter using Connolly surface function for a) AA-Eclipsed, b) AA'-serrated, c) AB staggered stacking modes of PBHP-TAPT-COF d) Nitrogen adsorption and desorption isotherms and e) Pore width analyses (NL-DFT) and f) Pore width analyses(QS-DFT) for PBHP-TAPT COF, (S_{BET} of PBHP-TAPT COF = 176 m²/g, nitrogen sorption at 77 K).

4. For structural simulation, there is an obvious difference between the experimental pattern and simulated patterns for AA-Serrated. The author should do the simulation again.

Simulated PXRD were obtained for structures fully optimized at the DFT-D3 level of theory at 0 K disregarding possible solvent molecules in the pores. This is a standard approach for finding the differences between various possible inter-layer arrangements. As is apparent from Supplementary Fig. 4, all calculated structures (AA-serrated, AA-eclipsed and AB-staggered) are showing 100, 110, 200 and 210 peaks at slightly higher 2θ values than experimental data. This is due to small underestimation of the UC vectors at the DFT level. We believe that new simulations are not likely to bring any improvement unless we include computationally much more expensive models that account for the (non-stoichiometric) presence of adsorbates and temperature in particular. Our assignment of the material structure is based on (i) energetic consideration that eliminates AB-staggered structure (that is more than 4 eV/UC higher than other two structure); (ii) simulated PXRD for AA-eclipsed structure does not resemble the experimental pattern, in particular, it shows two peaks around 2.5 and more complicated peak structures between 3 and 7 (2θ). We believe that these arguments allow us to conclude that experimental structure is dominantly AA (or AA') and the differences between experimental and simulated data are due to randomized inter-layer stacking (since there are 6 energetically equivalent interlayer vectors). Finally, irrespective of the fidelity of the DFT calculations, these DFT models served as the basis for a reasonable structural refinement of the observed PXRD profile (with a good fit of $R_{\text{wp}} = 3.91\%$), and we do not believe that there is more insight that can be gained from further DFT calculations.

5. The author used several techniques to verify the site for protonation. Did they try the solid-state NMR? There should some shift for the carbon of triazine ring during this protonation process.

We thank the reviewer for this suggestion. As we show in the course of the draft, simple de-sorption of HCl from our COF under RT and ambient conditions leads to a release of HCl vapours over short periods of time, and definitely over the timescale of a solid-state NMR experiment. Physi-de-sorption of HCl gas is neither compatible with (expensive) high-spin rotors nor the (even more expensive) solid-state NMR machine that we employ, hence, we cannot record these spectra. We would also like to point out that the site of protonation is verified unambiguously by orthogonal methods such as spectroscopy (FT-IR) and using test molecules.

Reviewer 2:

This manuscript reported the fabrication of a 2D COF from chemoresistant β -amino enone bridges and Lewis-basic triazine moieties. The obtained 2D COF material exhibits a rapid sensing response in the visible spectrum for volatile acid vapours. To date, a series of chemical reactions, synthetic methods and synthetic strategies have been developed, which greatly promoted the development of COFs. However, relative application of COFs materials in large-scale sensors is rare. This guarantees its publication in nature communications. Several issues should be addressed prior acceptance.

(1) COFs materials usually have uniform pore size distribution. Why does this material have such a wide range of pore size distributions?

Strictly speaking this first statement is not correct. While some structurally rigid COFs show a narrow distribution of pore sizes (e.g. our recently published Si-COF in Nature Chemistry, 2017, 9, 977, or Science, 2005, 310, 1166), there is a host of COFs that are sufficiently flexible or (in the case of layered 2D COFs) feature some degree of stacking disorder to show several contributions in PSD analysis (e.g. Nature Communications 2018, 9, Article number: 5234 or Chem. Commun., 2016,52, 2843).

The problem of PSD analysis is exacerbated by the limitations of NLDFT. While NLDFT is the preferred method for calculation of pore sizes in the materials community, we must remember that it is a method that works best for siliceous materials and zeolites: pore size analysis of carbons with heterogeneous surfaces and flexible pore structures remain difficult to predict with NLDFT. There is an excellent article that discusses the pros and cons of NLDFT and QSDFT elsewhere (J. Landers et al. / Colloids and Surfaces A: Physicochem. Eng. Aspects 437 (2013) 3–32). We chose to include results for PSD analysis from NLDFT and QSDFT methods in a re-worked Supplementary Fig. 8, especially since QSDFT is becoming increasingly more common for COF materials and heterogeneous carbons. (Carbon . 2014, 72, 47-56) , (Colloids and Surfaces A: Physicochem. Eng. Aspects, 201, 437 3-32), (Chem. Mater. 2016, 28, 626–631), (Langmuir 2006, 22, 11171-11179), (CrystEngComm, 2016, 18, 4295–4302) and (Nat Commun 2018, 9,2600). We have preformed pore size distribution analysis using QSDFT, and have obtained fundamentally better fits (see below).

We have also amended the discussion of PSD analysis on p. 9 as follows:

“Nitrogen sorption isotherms of **PBHP-TAPT COF** show Type I behaviour, indicative of pores, with a guest-accessible Brunauer–Emmett–Teller surface area (S_{BET}) of $176 \text{ m}^2 \text{ g}^{-1}$ (Supplementary Fig. 8d). Pore size distribution was calculated using quenched solid-state functional theory (QSDFT) (Supplementary Fig. 8e), and the largest contribution to pore volume stems from pores of diameter 3.9 nm which is close to the pore channel diameter of the refined structure of 3.6 nm (Supplementary Fig. 8a).”

Figure Modified in the SI :

Supplementary Fig. 8: Calculated pore diameter using Connolly surface function for a) AA-Eclipsed, b) AA'-serrated, c) AB staggered stacking modes of PBHP-TAPT-COF d) Nitrogen adsorption and desorption isotherms and e) Pore width analyses (NL-DFT) and f) Pore width analyses(QS-DFT) for PBHP-TAPT COF, (S_{BET} of PBHP-TAPT COF = 176 m²/g, nitrogen sorption at 77 K).

(2) The author said "the framework can also be fully deactivated by a mild thermal treatment (> 120 °C for 60 min) or in vacuum". What is the meaning of deactivation? Whether the material still realize the reversible activation by acid vapours after such a treatment? What is its structural change after such a treatment?

We apologise for not making this point clearer. In the context of the colorimetric study we describe as "activation" the exposure of the COF material to HCl (g) (which triggers the colour change to red) and as "deactivation" we describe the "regeneration" of the COF to its pristine state. "Deactivation" of the HCl-treated COF can be done by chemical means (i.e. by exposure to NH₃ (g)) or by physical means (i.e. triggering the desorption of HCl using temperature or vacuum). "Activation" does not change the structure of the material (as shown previously in Supplementary Fig. 14). We have further included a comprehensive UV-Vis and FT-IR study that verifies that "deactivation" (by chemical or physical means) does not alter the composition of the material (in Supplementary Fig. 20) as requested.

Further, we have re-worded the paragraph on p. 11 as follows.

"Intriguingly, the protonated, excited framework can also be fully regenerated by thermal treatment (> 120 °C for 60 min) or in vacuum. Structural integrity of PBHP-TAPT

COF after physical and chemical regeneration was confirmed with Fourier-transform infrared (FT-IR) spectra and solid state UV-Vis measurements (Supplementary Fig. 20).”

Figure Modified in the SI :

Supplementary Fig. 20: a) Photographs indicating regeneration of protonated PBHP-TAPT COF left to right: protonated sample, regenerated via NH₃ treatment, regenerated via heat treatment 120 °C 60 min, and regenerated under high vacuum on shlenk line for 24h at RT, b) FTIR spectra of samples regenerated using different methods, c) UV-Vis diffusive reflectance spectra of regenerated PBHP-TAPT COF, and d) UV-Vis diffusive reflectance spectra of reactivated PHBP-TAPT COF by HCl vapours.

(3) a) From the manuscript, it can be seen that the pristine PBHP-TAPT COF has a normalised conductivity of 1.32×10^{-8} S m. On protonation, the conductivity of activated PBHP-TAPT COF increases to 2.18×10^{-6} S m⁻¹. However, compared with the other materials the author listed, this conductivity is still low.

This is correct. The conductivity of (activated) PBHP-TAPT COF is comparable to other polymeric semiconductors (TzF/TzG with 1.6×10^{-6} S m⁻¹ in Adv. Mater. 2017, 29, 1703399) and undoped trans-polyacetylene ($\sim 10^{-6}$ S m⁻¹) or undoped cis-polyacetylene and PANI ($\sim 10^{-8}$ S m⁻¹).

It should be noted that in the context of sensor applications the 2-fold increase of conductivity which we detect upon activation of PBHP-TAPT COF with HCl vapours (from $\sim 10^{-8}$ to $\sim 10^{-6}$ S m⁻¹) is still significant and can be detected and exploited.

We hope that this addresses the reviewer's comment.

(4) b) What is the reason for the authors to choose 2Z,2'Z)-1,1'-(1,4-phenylene)bis(3-hydroxyprop-2-en-1-one) instead of 1,4-diacetylbenzene as the starting material? If selected 1,4-diacetylbenzene, the formed COF material should have a fully π -conjugated framework, which in turn will have a better conductivity.

This is an excellent suggestion for future experiments; our thanks for this.

The underlying design principle of PBHP-TAPT COF was to take advantage of the β -amino enones bridges along with the incorporation of triazine moieties to study donor-acceptor behaviour (which we found beneficial in the past for e.g. more efficient separation of photo-induced charge carriers; see: Angewandte Chemie International Edition 2018, DOI: 10.1002/anie.201809702).

We would nonetheless like to point out to our colleague that an increase/optimisation of conductivity is not a strict requirement for sensor applications.

Reviewer #3 (Remarks to the Author):

It is an excellent manuscript on the topic of "A real-time optical and electronic chemical sensor based on a β -amino enone linked, triazine-containing 2D covalent organic framework ". I think the m/s has been written well and the logic behind the m/s is nice. Hence I recommend acceptance per minor revision.

1. Once I look into the structure I felt that there is a high chance of an intramolecular hydrogen bonding [See *Angew. Chem. Int. Ed.*, 2013, 52, 13052–13056]. I was wondering if the authors saw some indication of such.

(We thank the reviewer for their positive appraisal and suggestions, and we would like to address point 1, 2, 3 and 4 together below, since they all relate to hydrogen bonding.)

2. Please see the N-H next to **b** in the figure 2b. If we replace that N-H with N-D via a D exchange I guess the possibility of this hydrogen bonding will more clear.

3. Similarly, an interlayer hydrogen bonding could be another possibility as well. See *J. Am. Chem. Soc.*, 2018, 140, 10941

4. The free C=O functionality and the very existence of that indicate such possibilities.

*Indeed, one would expect the presence of hydrogen bonding between the N-H and spatially close C=O groups as reported by Banerjee et.al. and e.g. in *ChemSusChem* 2017, 10, 921 – 929. And we see some evidence for intra-layer H-bonding as indicated in the FT-IR spectrum (Fig. 2a, peaks around 1570 to 1600 cm^{-1}).*

*To elucidate whether H-bonding could exist between neighbouring layers of our COF (inter-layer bonding), we examined the refined structure of the dominant, crystalline phase of the material, and the conceivable stacking modes obtained by DFT calculations. The maximum hydrogen acceptor distance was set to 3.3 Å and the minimum donor-hydrogen-acceptor angle was set to 90°. However, we did not find any conceivable angles for the interlayer H-bonding which are typically around 166° as reported in previous literature (*J. Mater. Chem. C*, 2017, 5, 2603–2610 and *J. Am. Chem. Soc.*, 2018, 140, 10941). We have included the results of this survey below for the benefit of the reviewer, but we did not add them as “negative results” to the manuscript or the SI.*

Figure (for reviewers only): Evaluation of hydrogen bonding for different stacking modes of **PBHP-TAPT COF** for a) AA eclipsed (as refined from PXRD data), b) AA' serrated (as found by DFT) and c) AB staggered (as found by DFT). Carbon (grey), nitrogen (blue), and hydrogen (white) atoms as shown as spheres. Conceivable hydrogen bonds are shown as dotted lines (teal). Potential hydrogen bonds were screened with a maximum hydrogen acceptor distance of 3.3 Å and a donor-hydrogen-acceptor angle between 180 and 90°.

It is apparent that neither in the observed nor in the conceivable (DFT) stacking arrangements any of the NH or CO groups of neighbouring layers are in the correct spatial arrangement to form inter-layer H-bonds. Although we do not categorically exclude the possibility of such inter-layer H-bonding for our material, I hope our colleague agrees that – based on the structural survey – there is no solid hypothesis that would warrant substantially more time-consuming NH/ND exchange experiments.

Reviewers' comments:

Reviewer #1 (Remarks to the Author):

I have gone through their response to comments very carefully. However, I think the authors didn't answer my question very well. Here are my response:

1. To be honest, I don't think the crystallinity of this COF is good enough. The authors compared the FWHM of the (100) peak with two reported systems, but this cannot strongly support their announcement. The (100) peak is not very sharp, especially compared to the reported β -amino enone linked COF (JACS, 2017, 139, 2421). In addition, there is a broad peak between 6-10 degrees. Moreover, the author claimed that they optimized the synthetic protocol, but that also doesn't guarantee they finally got the right condition. How many experiments did they perform? In my opinion, some groups usually sealed hundreds of tubes to get the final condition. The author should definitely improve the quality of the COF.

2. From the literature (JACS, 2017, 139, 2421), the reported COFs synthesized via Michael addition elimination showed a BET surface area of 505, 478, 258 and 252 m²/g. Therefore, this COF has a much lower surface area and I strongly believe this is related to the poor crystallinity. In addition, they claimed a higher N₂ BET surface area offers no tangible benefit for sensing application, so I would like to ask what is the advantage to synthesize the COF? Maybe the amorphous polymer can do the same thing.

3. For the PSD, the authors don't need to provide the NL-DFT model.

4. For the simulation, like I said before, there is an obvious deference between the experimental and simulated patterns, especially the 100 peak. The author claimed they used the standard approach to optimize the crystal structure, and they believe that new simulations are not likely to bring any improvement unless they include computationally much more expensive models. This is definitely not a good reason! For simulation of COF structure, the most important part is that the simulated main peak (usually 100) should have the same position with the experimental pattern.

5. The author claimed the reason for this pH responsive process is related to the protonation and deprotonation of triazine core. However, from the literature (JACS, 2017, 139, 2421), it sounds the secondary amine site can also be protonated. So I asked them to take the solid-state NMR to prove it. They think it is very difficult and don't want to do it. Come on! This experiment is definitely possible, for example, they can seal the NMR tube that was filled with HCl.

6. I should point out a paper entitled "A gaseous hydrogen chloride chemosensor based on a 2D covalent organic framework" published recently (Chem. Commun., 2019, 55, 4550). This COF exhibited an obvious color change from yellow to red upon exposure to HCl gas with a response time of less than 1 second.

Reviewer #2 (Remarks to the Author):

The authors have explained the problem I raised very well. I have no doubt about this manuscript. The current version can be published in Nature Communications.

Reviewer #3 (Remarks to the Author):

I am satisfied with the changes and the m/s could be accepted as it is.

In response to reviewer 1:

- 1. To be honest, I don't think the crystallinity of this COF is good enough. The authors compared the FWHM of the (100) peak with two reported systems, but this cannot strongly support their announcement. The (100) peak is not very sharp, especially compared to the reported β -amino enone linked COF (JACS, 2017, 139, 2421). In addition, there is a broad peak between 6-10 degrees. Moreover, the author claimed that they optimized the synthetic protocol, but that also doesn't guarantee they finally got the right condition. How many experiments did they perform? In my opinion, some groups usually sealed hundreds of tubes to get the final condition. The author should definitely improve the quality of the COF.**

A claim of “crystallinity [...] is not good enough” is not a helpful or scientific statement. We have provided a comprehensive overview of an actual measure for crystallinity (FWHM analysis of COFs), and we find that the (100) peak of our reported PBHP-TAPT COF has a FWHM of 0.4° , while other COFs have similar FWHM of $0.4-0.7^\circ$ (“Tunable near white-emissive two-dimensional covalent organic frameworks” Nat. Commun. 2018, 9, 2335) and $0.39-0.6^\circ$ (“Stable, crystalline, porous, covalent organic frameworks as a platform for chiral organocatalysts” Nature Chemistry, 2015, 7, 905–912). The reported 3'PD and 2TPA samples that reviewer #1 refers to (J. Am. Chem. Soc. 2017, 139, 2421-2427) have a theoretical pore size of around 4.08 and 3.55 nm – close to the pore size we report for our material. From the supporting information section, we see two-fold: the (100) feature for 3'PD (Figure S10) is almost entirely overlapping with the primary beam making FWHM analysis impossible, and 2TPA (Figure S11) does not show any discernible reflection peaks at all.

Again, it is unclear what point the reviewer is trying to make. In particular the reviewer's comments regarding the “quantity” of performed experiments in our work are not constructive or helpful – but to address the reviewer's comment on “How many experiments did they perform? In my opinion, some groups usually sealed hundreds of tubes to get the final condition.”: we have performed a sufficient number of experiments to ensure that our reported results are reproducible – that many.

Figure S10: Comparison of PXRD pattern of as-synthesized 3'PD (top) and corresponding simulated patterns for eclipsed and staggered stacking models (molecular models are shown on the right).

Figure S11: PXRD pattern of 2tpa as-synthesized (left) and a molecular model of its eclipsed stacking.

https://pubs.acs.org/doi/suppl/10.1021/jacs.6b12005/suppl_file/ja6b12005_si_001.pdf

2. From the literature (JACS, 2017, 139, 2421), the reported COFs synthesized via Michael addition elimination showed a BET surface area of 505, 478, 258 and 252 m²/g. Therefore, this COF has a much lower surface area and I strongly believe this is related to the poor crystallinity. In addition, they claimed a higher N₂ BET surface area offers no tangible benefit for sensing application, so I would like to ask what is the advantage to synthesize the COF? Maybe the amorphous polymer can do the same thing.

The authors from the above reference (J. Am. Chem. Soc. 2017, 139, 2421-2427) clearly state that the gas-accessible surface areas obtained by degassing of their materials in vacuum are comparatively poor. “However, enlarging the unit cell by 0.75 nm in 3BD unexpectedly led to a slightly lower BET surface area of 478 m² /g, and even much lower porosity was revealed by the COFs 3'PD and 2TPA. We also note that when the activation was carried out by simple vacuum drying (120 °C), ca. twice lower surface areas were measured for all COFs” (page 2424, 1st paragraph).” Only CO₂ activation afforded higher surface areas. More specifically, for 3PD and 3BD samples the recorded surface areas were 152 and 109 m²/g, respectively (Table S2: Pore distributions of COFs). They did not record any surface area values for 3'PD and 2TPA at all (see below).

Figure S11: PXRD pattern of **2tpa** as-synthesized (left) and a molecular model of its eclipsed stacking.

Table S2: Pore distributions of COFs

Sample	BET (N ₂ adsorption) (m ² /g)(supercritical CO ₂ activated)	Pore distributions (nm)	Theoretical pore size (DFT) (nm)	BET (N ₂ adsorption) (vacuum activated) (m ² /g)	BET (CO ₂ adsorption) (m ² /g)	CO ₂ uptake (mg/g) at 0.3 bar (273 k)
3PD	505	1.27	2.54	152	180	117
3BD	478	1.33	3.29	109	111	78
3'PD	258	3.19 and 4.52	4.08	---	---	
2TPA	252	9.70	3.55	---	---	

Hence, it is unclear what point the reviewer is trying to make here. The reason for the comparatively low surface area of our material is the drying/degassing protocol (as reported previously) and not incomplete condensation, as the reviewer is trying to suggest. Complete condensation of our network is evident from orthogonal techniques like elemental analysis, EDX and solid-state NMR.

In reference to the second part of the reviewer's comment: "I would like to ask what is the advantage to synthesize the COF? Maybe the amorphous polymer can do the same thing." The advantage of making the COF is that we have a clearer understanding of make-up and positioning of functional groups in the material than in an amorphous polymer. Whether the amorphous material could or could not achieve the "same thing" is a hypothetical discussion: polymorphism between COFs and their amorphous analogues has been achieved only once in literature – by us ("Tuning the Porosity and Photocatalytic Performance of Triazine-Based Graphdiyne Polymers through Polymorphism" ChemSusChem 2019, 12, 194.).

3. For the PSD, the authors don't need to provide the NL-DFT model.

We disagree with the reviewer. NL-DFT is a standard method for calculating PSD, and although it is being superseded by QS-DFT, we will continue reporting a standard.

4. For the simulation, like I said before, there is an obvious deference between the experimental and simulated patterns, especially the 100 peak. The author claimed they used the standard approach to optimize the crystal structure, and they believe that new simulations are not likely to bring any improvement unless they include computationally much more expensive models. This is definitely not a good reason! For simulation of COF structure, the most important part is that the simulated main peak (usually 100) should have the same position with the experimental pattern.

Simulations of PXRD patterns in Supplementary Fig. 4 show some discrepancy because they do not account for instrumental zero-point shift – as is an overwhelming standard for reporting DFT data together with instrumental PXRD profiles. Since this is confusing the colleague, we have revised Suppl. Fig. 4 to account for the zero-point shift that we obtained via Pawley refinement.

Again, we have to state that we employ state-of-the-art DFT methods, and suggesting further delays by “computationally much more expensive models” is neither helpful nor constructive.

Supplementary Fig. 4: Comparison of PXRD pattern of PBHP-TAPT COF (black) and corresponding simulated patterns for AA-Eclipsed, AA-Serrated and AB-Staggered (accounting for zero-point instrumental shift obtained from Pawley refinement).

5. The author claimed the reason for this pH responsive process is related to the protonation and deprotonation of triazine core. However, from the literature (JACS, 2017, 139, 2421), it sounds the secondary amine site can also be protonated. So I asked them to take the solid-state NMR to prove it. They think it is very difficult and don't want to do it. Come on! This experiment is definitely possible, for example, they can seal the NMR tube that was filled with HCl.

For clarification: we are not talking about pH responses at all. We claim that it is the protonation and deprotonation of the triazine core that has the major contribution to the observed red-shift in the UV/Vis.

The reviewer's exclamation that "[they] don't want to do it. Come on!" is not very helpful, since performing these solid-state NMR experiments necessitates the procurement of a special solid-state NMR rotor with an air-tight O-ring. "[sealing] the NMR tube that was

filled with HCl” is not an option – we are not dealing with glass NMR tubes (these are all solid-state NMR experiments), and we do not use aq. HCl solution but HCl gas. The reviewer is confusing methods and set-ups here.

However, we have included these experiments in “Section 24: Solid-state ^{13}C NMR of pristine and protonated PBHP-TAPT COF”, and we have added the findings to the SI as follows:

In order to further confirm the protonation site, we performed solid-state ^{13}C CP-MAS NMR on the HCl-activated **PBHP-TAPT COF**. 0.2 g of pristine sample was protonated using a steady stream of HCl gas for 10 s, as in all previous protonation experiments. ^{13}C CP-MAS solid-state NMR spectra of **PBHP-TAPT COF** were recorded in 3.2 mm rotors at 13 kHz, and protonated **PBHP-TAPT COF** spectra were obtained in 4 mm rotors at 10 kHz. The ^{13}C signals were recorded for 12 h. It should be noted, that the 4 mm rotor was not absolutely airtight, as we observed a color change of the protonated **PBHP-TAPT COF** from deep red to dark orange over the course of 12 h.

Upon protonation, most peaks experience an upfield shift, with the notable exceptions of the aryl sp^2 carbon environment (e) and the keto-enamine carbon (a). This corresponds best to the scenario that the protonation site is preferentially at the ring-nitrogen of the triazine sub-unit (see Supplementary Figure 25).

Supplementary Figure 1: Solid-state ^{13}C CP-MAS spectra of pristine and protonated PBHP-TAPT COF shown in black and in red, respectively.

The figure below shows the predicted chemical shifts of carbon signals upon protonation; all three different possibilities were considered and are compared in the table below: 1) protonation only at the triazine, 2) protonation at the triazine and keto-enamine bridge, and 3) protonation only at the KE. Based on the comparisons the ^{13}C NMR suggests that the triazine core is the preferred protonation site.

Predicted ^{13}C NMR:

Carbon nuclei	Pristine Observed (ppm)	Protonated Observed (ppm)	Pristine Predicted ^c (ppm)	Triazine protonated ^c (ppm)	Triazine + KE protonated ^c (ppm)	Only KE protonated ^c (ppm)
a	95	93	94	94	78	78
b	147	140	146	146	157	157
c	189	186	189	189	177	177
d	139	131	137	137	135	135
e	129	130	128	128	127	127
f	140	135	139	139	143	143
g	114	112	114	114	125	125
h	128	126	128	128	128	127
i	127	121	124	116	125	134
j	179	165	172	162	162	172

Note: ^c= Predicted NMR signals
Blue = Downfield shift
Red = Upfield shift
Black = No change

Supplementary Figure 2: Observed and predicted ^{13}C signals of several protonation sites.

Furthermore, we have included the study of further two model compounds 1,3,5-tris(4-aminophenyl)benzene (TAP) and 1,3,5-tris-(4-aminophenyl) triazine (TAPT) (Supplementary Section 18) that clearly shows that a red-shift of the UV/Vis absorption edge is only visible for compounds that contain the triazine core.

Supplementary Figure 3: Solid-state UV/Vis measurements of the model compounds: 1,3,5-tris(4-aminophenyl)benzene (TAP-Amine); 1,3,5-tris-(4-aminophenyl) triazine (TAPT-Amine); and 2,4,6-triphenyl-1,3,5-triazine (TPT). Protonation was performed using a stream of HCl (g) for 10 s.

In Section 19, the protonation of **PBHP-TAPT COF** has been investigated computationally, using the same models and methods as described above in computational details in the main text. Protonation sites on the N atom of triazine and on the N atom of the keto-enamine linker were considered. The DFT results show that the protonation on triazine is preferred by 70 kJ mol^{-1} over protonation on the keto-enamine linker. The proton added on the N atom of triazine is not involved in any H-bonding. On the contrary, the proton added to the N atom of the keto-enamine linker is stabilized by the formation of the H-bond with the O atom of the linker at the adjacent layer. At the minimum energy structure, the proton is shifted from N to O atom of the keto-enamine linker.

Supplementary Figure 4: The protonation of (a) the triazine and (b) the bridge of the **PBHP-TAPT COF**. Position of added proton is shown within the red circle. Color scheme: H (white), C (brown), N (blue) and O (red)

Supplementary Figure 5 : The cluster model for protonation of the linker; view from different directions; grey, blue, red and white balls represent C, N, O and H, respectively.

We carried out the same calculations using cluster model consisting of triazine-linker. While the relative energies of protonated connector and linker was similar (77 kJ mol^{-1} in favor of triazine protonation), the minimum energy structure was different: H is primarily bound to the N atom and it

forms an H-bond with the O atom which is moved into the optimal position for H-bonding (a planar O-C-C-C-N-H ring. Such rearrangement requires rotation along two backbone C-N bonds which is forbidden in the 2D PBHP-TAPT COF (adjacent connectors would not be in the same plane). Thus, the rigidity of 2D **PBHP-TAPT COF** leads to the proton transfer from N to O atom of the linker.

We have also expanded the details in the Fourier-transform infrared spectroscopic study that verifies preferred protonation of the triazine moiety.

Section 20: FT-IR Spectra of PBHP-TAPT COF before and after HCl gas exposure

Supplementary Figure 6: Fourier-transform infrared (FT-IR) spectroscopic study on protonation and deprotonation of PBHP-TAPT COF, pristine PBHP-TAPT COF, protonated PBHP-COF (activation with HCl vapours) and regeneration by NH₃ for the conformation of protonation site and characteristics of triazine and keto-enamine regions affected due to protonation by HCl gas.

In conclusion we find that although electronic delocalisation along the π -aromatic backbone of the framework contribute to the magnitude of the UV/Vis shift, it is clear from these experiments and calculations that the triazine moiety of **PBHP-TAPT COF** is the preferred protonation site– as claimed previously.

6. I should point out a paper entitled “A gaseous hydrogen chloride chemosensor based on a 2D covalent organic framework” published recently (Chem. Commun., 2019, 55, 4550). This COF exhibited an obvious color change from yellow to red upon exposure to HCl gas with a response time of less than 1 second.

Yes, we are well aware of that paper from Feb/Mar 2019, and we congratulate the authors of that paper who published a completely different system (based on imine bonds) with a similar optical effect in the time after our draft went to review at Nature Communications (Jan 2019).

REVIEWERS' COMMENTS:

Reviewer #1 (Remarks to the Author):

I went through their response very carefully. To be honest, I am quite frustrated as the author don't want to improve the quality of the COF. Here are my comments:

1. The PXRD pattern of this COF shows several peaks, which can prove the presence of a long-range ordered structure to some extent. Unfortunately, this PXRD pattern had a strong background and showed a broad peak between 10- 15 degree, which strongly indicates the existence of amorphous polymers. In addition, by comparing with the reported system (Nature Chemistry, 2015, 7, 905; Nat. Commun. 2018, 9, 2335), the author used FWHM analysis to support their statement. However, I would like to say they should not ignore that these reported system don't have strong background and the baseline is quite flat. Therefore, I don't think the crystallinity of this COF is good enough for publication and the author should definitely improve the quality of the COF.

2. The observed surface area (176 m²/g) is far from the expected Connolly surface area (2399 m²/g). The author supposed it should be ascribed to the stacking defects or interdigitation of neighbouring layers. This is possible, but not enough!!! Like I mentioned, the formation of amorphous polymers during the COF synthesis can also result in the low surface area. Again, the author should try more reaction conditions and improve the crystallinity of this COF. Indeed, the surface area of this COF is very close to the literature (JACS, 2017, 139, 2421), but this does not mean their system is good enough to publish in a highly standard journal. First, the reference reported the synthesis of a COF via Michael Addition-Elimination for the first time, which is quite novel. Second, the reference has been published for over two years. During this period, several new reactions has been used to construct the COFs and two landmark papers about single crystal of COFs has been reported. In other words, this is a very complete area and the community is paying much attention on the quality of COF now. Therefore, I don't think this manuscript can meet the highly standard of the journal Nature Communications.

Reviewer #2 (Remarks to the Author):

i) how serious you think the concern about the crystallinity are.

Usually, the covalent organic frameworks (COFs) materials are believed to be one type of materials, which has well-defined crystalline porous structures together with tailored functionalities. These offering them superior properties in diverse applications. So, the properties of COFs are close related with its crystallinity, BET and porous structure. In our previous experiments, we found when the quality of COFs is not good enough, they still exhibit moderate XRD, but along with poor BET and wide porous size distribution. From this point, I agree with this reviewer, the quality of the present COF is not good enough. Modulating the experimental condition would improve its quality and in turn function.

ii) If the authors could clearly answer the question regarding the protonation of the triazine and the secondary amine site in the COF.

From the experimental and computational data, the authors could clearly point out the triazine moiety of PBHP-TAPT COF is the preferred protonation site.

Point-by-point response to reviewer comments NCOMMS-19-04469B

Reviewer #1 (Remarks to the Author):

I went through their response very carefully. To be honest, I am quite frustrated as the author don't want to improve the quality of the COF. Here are my comments:

1. The PXRD pattern of this COF shows several peaks, which can prove the presence of a long-range ordered structure to some extent. Unfortunately, this PXRD pattern had a strong background and showed a broad peak between 10-15 degree, which strongly indicates the existence of amorphous polymers. In addition, by comparing with the reported system (Nature Chemistry, 2015, 7, 905; Nat. Commun. 2018, 9, 2335), the author used FWHM analysis to support their statement. However, I would like to say they should not ignore that these reported system don't have strong background and the baseline is quite flat. Therefore, I don't think the crystallinity of this COF is good enough for publication and the author should definitely improve the quality of the COF.

Dr. Michael J. Bojdys (MRSC)

Berlin, 18/06/2019

Functional Nanomaterials
Department of Chemistry
Humboldt-Universität zu Berlin
Brook-Taylor-Str. 2
12489 Berlin
Germany

e-mail: m.j.bojdys.02@cantab.net
tel: (+49) 1774818190

We are sorry to see that reviewer #1 has not changed their objections and persists with their subjective assessment that the "crystallinity of [our] COF is [not] good enough". The as-presented COF material is the result of many optimisation steps (in terms of temperature, time and solvents) which are all well-documented in the Supporting Information section. We have also made reviewer #1 aware that the peak assignments are confirmed by Pawley fitting, state-of-the-art computational simulations at DFT level and a comprehensive comparison of crystallinity with other COF systems in literature which verifies the fidelity of the here-presented system. The new discussion of the base-line that reviewer #1 is trying to bring forward is "base-less" given the fidelity of the Pawley fit.

2. The observed surface area (176 m²/g) is far from the expected Connolly surface area (2399 m²/g). The author supposed it should be ascribed to the stacking defects or interdigitation of neighbouring layers. This is possible, but not enough!!! Like I mentioned, the formation of amorphous polymers during the COF synthesis can also result in the low surface area. Again, the author should try more reaction conditions and improve the crystallinity of this COF.

Indeed, the surface area of this COF is very close to the literature (JACS, 2017, 139, 2421), but this does not mean their system is good enough to publish in a highly standard journal. First, the reference reported the synthesis of a COF via Michael Addition-Elimination for the first time, which is quite novel. Second, the reference has been published for over two years. During this period, several new reactions has been used to construct the COFs and two landmark papers about single crystal of COFs has been reported. In other words, this is a very complete area and the community is paying much attention on the quality of COF now. Therefore, I don't think this manuscript can meet the high standard of the journal Nature Communications.

Again, we reiterate that the quality of our COF material is more than adequate, and the phenomena of pore-closure are described in detail within the text as well as documented in previous instances of reported COF materials which are adequately referenced. The sheer fact that the here-presented COF material is described not only by orthogonal methods in most (if not all) of its complexity, and that it shows an unprecedented chemical and electrical response to gas species in real-time (a first for any COF material) qualifies the reporting of these findings in Nature Communications.

Reviewer #2 (Remarks to the Author):

i) how serious you think the concern about the crystallinity are.

Usually, the covalent organic frameworks (COFs) materials are believed to be one type of materials, which has well-defined crystalline porous structures together with tailored functionalities. These offering them superior properties in diverse applications. So, the properties of COFs are close related with its crystallinity, BET and porous structure. In our previous experiments, we found when the quality of COFs is not good enough, they still exhibit moderate XRD, but along with poor BET and wide porous size distribution. From this point, I agree with this reviewer, the quality of the present COF is not good enough. Modulating the experimental condition would improve its quality and in turn function.

We assure the colleague that the quality of the COF can not be improved further on any realistic time-scale. The here-reported COF material is the result of many optimisation steps (in terms of temperature, time and solvents) which are all well-documented in the Supporting Information section. We have also made the point in a previous revision step that the peak assignments are confirmed by Pawley fitting, state-of-the-art computational simulations at DFT level and a comprehensive comparison of crystallinity with other COF systems in literature which verifies the fidelity of the here-presented system.

ii) If the authors could clearly answer the question regarding the protonation of the triazine and the secondary amine site in the COF.

From the experimental and computational data, the authors could clearly point out the triazine moiety of PBHP-TAPT COF is the preferred protonation site.

We verify that the preferred protonation site is indeed the triazine moiety via orthogonal methods: (a) ^{13}C CP-MAS spectroscopy, (b) experiments with model compounds, (c) FT-IR spectroscopy, and (d) calculations at DFT level (sections 19 and 20 in the Supporting Information section).